# Position: Formal Mathematical Reasoning—A New Frontier in AI

Kaiyu Yang [1]   Gabriel Poesia [2]   Jingxuan He [3]   Wenda Li [4]   Kristin Lauter [1]   Swarat Chaudhuri [5]   Dawn Song [3]

## Abstract

AI for Mathematics (AI4Math) is intellectually intriguing and is crucial for AI-driven system design and verification. Extensive efforts on AI4Math have mirrored techniques in NLP, in particular, training large language models on carefully curated math datasets in text form. As a complementary yet less explored avenue, *formal mathematical reasoning* is grounded in formal systems such as proof assistants, which can verify the correctness of reasoning and provide automatic feedback. This position paper advocates formal mathematical reasoning as an indispensable component in future AI for math, formal verification, and verifiable generation. We summarize existing progress, discuss open challenges, and envision critical milestones to measure future success.[1]

## 1. Introduction

Mathematical reasoning has been important for AI from its early days (Newell & Simon, 1956) to the era of modern large language models (LLMs). It serves as a proxy for reasoning and planning tasks and plays a fundamental role in quantitative disciplines. AI4Math has the potential to revolutionize AI for science, engineering, and beyond.

Substantial research in AI4Math has focused on math LLMs. A common approach is to continue pretraining LLMs on mathematical data from the Web and finetune on curated math problems with detailed solutions. We call this the "informal" approach to distinguish it from the formal approach that will be introduced later. Math LLMs have a simple recipe, but the secret sauce is data curation. Carefully curated training data plus inference-time techniques, such as chain-of-thought (Wei et al., 2022), have led to re-

markable success on benchmarks such as GSM8K (Cobbe et al., 2021) and MATH (Hendrycks et al., 2021). However, the success has been mostly limited to high school math, raising a key question: *How far can we go by scaling up the informal approach? Will it solve research-level mathematics problems?*

The informal approach faces challenges in dealing with advanced mathematics. First, high-quality training data is inherently scarce in advanced mathematics. Second, solutions to advanced problems are not limited to numbers but may include chains of intricate reasoning steps, such as proofs. LLMs are notorious for hallucinating seemingly valid reasoning steps, making it challenging to verify the correctness of model output or collect useful learning feedback. These challenges cannot be resolved by merely scaling up training. Researchers are exploring alternatives, such as scaling up inference (OpenAI, 2024), learning to reason via reinforcement learning (Guo et al., 2025), and neural verifiers (Cobbe et al., 2021). While these techniques have shown promise, we argue that neural networks alone are insufficient (Sec. 6). **A critical yet underexplored piece of the puzzle is formal mathematical reasoning. Combining formal reasoning with LLMs can significantly advance AI4Math, unlocking its applications in AI-driven formal verification and verifiable generation.**

We consider formal mathematical reasoning broadly as *mathematical reasoning grounded in formal systems*, including but not limited to higher-order logic (Nipkow et al., 2002), dependent type theory (Barras et al., 1997), and computer programs annotated with formal specifications (Leino, 2010). Formal systems provide environments that can verify the model's reasoning and provide automatic feedback. The feedback can mitigate data scarcity; also, such systems enable rigorous test-time checks that resist hallucination. In contrast, *informal mathematics* refers to math commonly found in textbooks and research papers. It interleaves natural language with symbols (e.g., LaTeX), but these symbols do not have a self-contained semantics, instead relying on informal text to convey significant parts of their meaning.

AlphaProof (Google DeepMind, 2024) and AlphaGeometry (Trinh et al., 2024) are two successful examples of this idea, leading to unprecedented performance in the International Mathematical Olympiad (IMO). They build on a

[1]Meta FAIR [2]Stanford University [3]UC Berkeley [4]University of Edinburgh [5]UT Austin. Correspondence to: Kaiyu Yang <kaiyuy@meta.com>, Dawn Song <dawnsong@cs.berkeley.edu>.

*Proceedings of the 42$^{nd}$ International Conference on Machine Learning*, Vancouver, Canada. PMLR 267, 2025. Copyright 2025 by the author(s).

[1]An extended version of this paper is available at https://arxiv.org/abs/2412.16075

broad literature on the synergistic use of formal methods and machine learning in mathematical tasks (Kaliszyk et al., 2018; Gauthier et al., 2021), including neural theorem proving, i.e., generating formal proofs given formal theorem statements, and autoformalization, i.e., automatically translating informal mathematics into formal mathematics. The advent of LLMs has significantly accelerated research in this area. For example, autoformalization was long hampered by the lack of aligned informal-formal data for finetuning. LLMs can mitigate this problem by either synthesizing the data (Jiang et al., 2024) or performing autoformalization without finetuning (Wu et al., 2022). LLMs are also powerful tools for theorem proving; in particular, recent approaches have used them to predict proof steps and fix buggy proofs (Thakur et al., 2024; First et al., 2023).

The research infrastructure around LLMs and formal reasoning is rapidly maturing. Lean (de Moura et al., 2015)—a language for writing formal proofs—has gained popularity among mathematicians. Multiple frameworks can support the interaction between LLMs and Lean (Yang et al., 2023; Aniva et al., 2024; Thakur et al., 2024). LLMs can assist humans in writing formal proofs (Song et al., 2025), potentially initiating a data flywheel where growing human-written formal math data leads to more capable LLMs, which in turn eases the creation of more data.

Emerging opportunities have led to booming research activities in AI for formal mathematical reasoning. The number of publications in this field has almost doubled annually in 2023 and 2024 (Li et al., 2024b). AlphaProof leveraged formal reasoning to become the first AI to achieve the silver medal level in IMO, followed by strong results from ByteDance and Harmonic in IMO 2025.[2] Developments in this field also have immediate applications in formal verification (Klein et al., 2009). While formal verification can lead to software and hardware systems that are exceedingly robust and secure, it has historically been too costly to deploy in all but the most safety-critical applications. AI can drastically reduce this cost by automating the formalization and proof effort needed to formally certify complex systems. This can lead to a future in which mass-produced software and hardware systems are far more robust than they are today.

We believe AI-based formal mathematical reasoning has reached an inflection point, with significant progress in the near future. This position paper maps out open challenges in data and algorithms and potential routes for future progress. It is not meant to be a comprehensive survey but to provide perspectives on where the field may go next and call on the community to unite to accelerate the progress.

---

[2]Google DeepMind, ByteDance, and Harmonic only announced their IMO results on blogs and X (formerly Twitter) without further details.

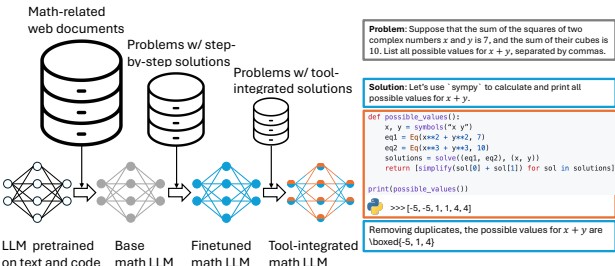

*Figure 1.* A typical math LLM.

## 2. AI4Math and the Formal Turn

After discussing the limitations of the informal approach, we introduce formal reasoning as a promising path.

### 2.1. State-of-the-art Math LLMs and Their Limitations

**A Case Study of NuminaMath.** NuminaMath (Fleureau et al., 2024) won the first AIMO Progress Prize in July 2024 and is an excellent example of modern math LLMs, as it encompasses many key ingredients:

1. *Math pretraining* (Fig. 1 *Left*): The base math LLM results from continually pretraining a generic LLM on mathematical Web documents. NuminaMath adopted DeepSeekMath-Base 7B (Shao et al., 2024) as the base math LLM, which was trained on high-quality mathematical documents retrieved from Common Crawl through a carefully engineered data pipeline that combined automatic filtering and manual annotation.

2. *Finetuning on step-by-step solutions* (Fig. 1 *Middle*): To align the model with problem solving, one can finetune it on a carefully curated dataset of math problems with detailed, step-by-step solutions. NuminaMath collected 860K problems and solutions covering high school and competition math (Li et al., 2024a).

3. *Tool-integrated reasoning* (Fig. 1 *Right*): Finetuned math LLMs may still struggle with precise calculation (e.g., $162 \times 731$) and symbol manipulation. A simple solution is to outsource these operations to external tools such as SymPy (Meurer et al., 2017). NuminaMath performs tool-integrated reasoning that interleaves reasoning in natural language with tool invocation in Python. The key is, again, *data*. They follow ToRA (Gou et al., 2024b) and MuMath-Code (Yin et al., 2024) to collect a dataset of math problems with solutions in the form of natural language interleaved with tool invocation trajectories.

**Data Scarcity.** Training data plays a pivotal role throughout all ingredients of the informal approach, limiting its

success to domains with abundant high-quality data, such as pre-college math. It is difficult to extend the approach to data-scarce domains such as advanced mathematics. Advanced mathematics is important for AI-driven scientific discovery, as it serves as the foundation of numerous scientific disciplines (e.g., climate modeling depends on partial differential equations). Moreover, the long-term goal of developing human-level AI mathematicians requires AI to handle novel aspects of mathematics. Novelty, by definition, implies difficulty in collecting in-distribution training data.

**Lack of Correctness Verifiability.** Another challenge lies in evaluation. Existing math LLMs are evaluated on benchmarks like MATH because many pre-college problems have numeric solutions that can be checked easily. For advanced mathematics, however, restricting to numeric solutions would deviate from common practice (Glazer et al., 2024), as it frequently deals with abstract conjectures and proofs. Verifying proofs can be a daunting task, even for experienced mathematicians (Klarreich, 2018). The situation becomes even more complicated when LLMs are used to generate proofs, as they are known to hallucinate plausibly.

## 2.2. AI for Formal Mathematical Reasoning

**From Informal to Formal.** *Formal mathematics* addresses the challenges in data and evaluation, potentially enabling AI to tackle advanced mathematics. In this paper, it refers to mathematics grounded in formal systems, which have a syntax for well-formed formulas and can perform reasoning by manipulating formulas according to rules. Examples of formal systems include higher-order logic (Gordon, 2000) and dependent type theory (Martin-Löf & Sambin, 1984). They are used not only in math but to express computer programs and reason about semantics (Howard, 1980).

Formal systems are useful environments for AI to learn math. They guarantee the soundness of the reasoning, provide automatic feedback, and check if the goal has been achieved. This is crucial to addressing existing challenges in data scarcity and evaluation. Automatic feedback can serve as learning signals and alleviate the need for *human-created* training data. Rigorous proof verification can evaluate the model's reasoning without worrying about hallucination.

**Proof Assistants and Lean.** An important type of formal system is *proof assistants*. These are software tools that enable humans to write formal proofs about mathematics or verified software. Common examples of proof assistants include Coq (Barras et al., 1997), Isabelle (Nipkow et al., 2002), and Lean (Moura & Ullrich, 2021). They have different logical foundations but similar user interfaces. For simplicity, we will use Lean as an example.

Fig. 2 demonstrates how Lean is used to formalize mathe-

matics. At its core, it is a programming language for writing not only conventional programs but also mathematical definitions, theorems, and proofs. Fig. 2 (*Middle*) is a Lean file. After defining natural numbers (Nat) and addition (add), it states and proves the theorem add_zero ($\forall n \in \mathbb{N}, 0 + n = n$). Lean can automatically check if the proof is correct with respect to the theorem statement.

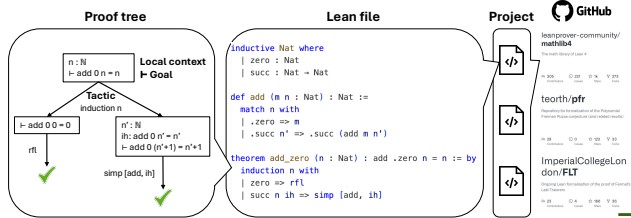

*Figure 2.* Formalizing math using Lean (de Moura et al., 2015).

Theorem proving in Lean is an interactive process (Fig. 2 *Left*). It begins with the statement as the initial goal, and the user enters a proof step, known as a "tactic". Lean executes the tactic, transforming the goal into a list of subgoals. The user then inspects the new goals and enters new tactics, repeating this process until there are no goals left. This process implicitly defines a proof tree whose nodes are goals and edges are tactics. The user plays a key role here. While proof assistants like Lean were designed with human users in mind, in formal mathematical reasoning, the user can also be AI or human mathematicians in collaboration with AI (Buzzard, 2024; Tao, 2024).

Formalizing mathematics using Lean is like developing software (Fig. 2 *Right*). Files are organized into larger code units such as libraries and projects, which can be open-sourced on GitHub and reused by other projects. For example, the formalization of cutting-edge research often builds upon the basic concepts formalized in mathlib (Mathlib community, 2020): Lean's general-purpose mathematical library.

**AI Meets Formal Mathematics.** Integrating AI with proof assistants such as Lean can be mutually beneficial. Proof assistants provide data and environments for AI to learn math, whereas AI enhances the user experience of proof assistants, e.g., by automating simple proofs. Fig. 3 illustrates common tasks at this intersection: Given informal mathematics from textbooks or papers, *autoformalization* automatically translates it into formal theorems and proofs (Fig. 4). Given theorem statements, *theorem proving* aims to generate formal proofs. In addition to the statement, a theorem prover may have access to a large library of existing definitions and lemmas, such as mathlib, and can select useful definitions and lemmas from the library. Furthermore, AI for autoformalization and theorem proving can lead to new theorems and/or proofs that can enrich the library.

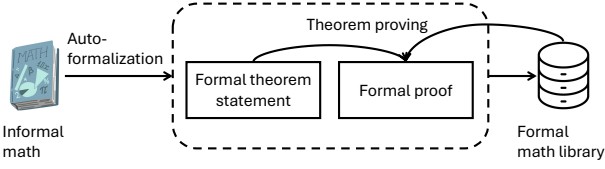

Figure 3. AI for formal mathematical reasoning in proof assistants.

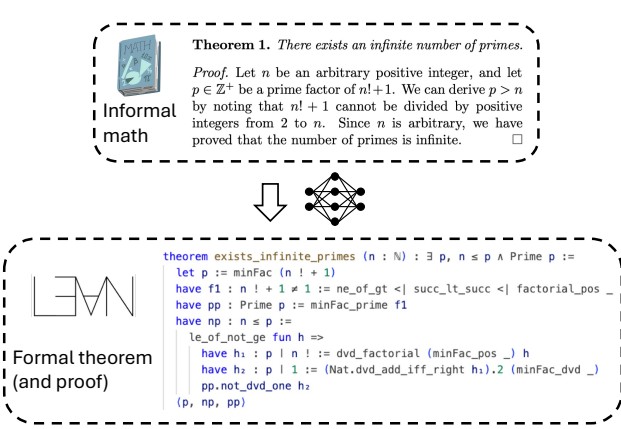

Figure 4. Autoformalization: from informal to formal.

Fig. 5 is a common architecture for neural theorem provers, which combines tactic generation and proof search. Given the current goal, a neural network generates suggestions for the next tactic. The network is often trained on human-written proofs and can be finetuned using reinforcement learning. The generated tactics are assembled into a complete proof through repeated sampling or tree search.

**Synergies with Natural Language and System Design.**
The formal and informal approaches are not mutually exclusive, nor should formal reasoning entirely supplant the informal. Instead, they can complement each other to enable complex reasoning that is both general and rigorous, e.g., integrating autoformalization with theorem proving to solve problems formulated in natural language (Zhou et al., 2024a). We refer to the combination of formal and informal reasoning as *verified reasoning in natural language*.

Moreover, formal mathematics has direct applications in the verification of software and hardware systems (Leroy, 2009). Here, one specifies the correctness/security requirements as formal statements and uses theorem proving to establish that the system satisfies its requirements. AI-driven autoformalization and theorem proving can potentially facilitate this process, significantly reducing the costs.

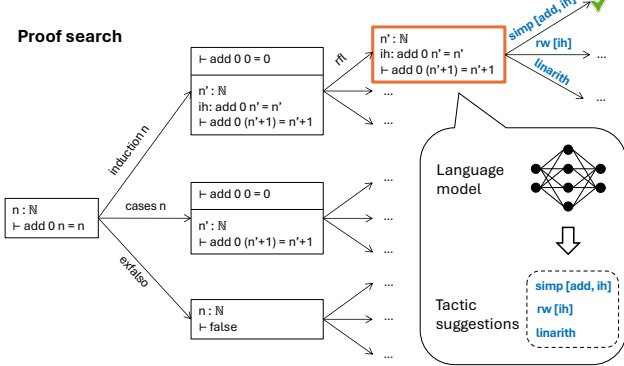

Figure 5. A common architecture for neural theorem proving.

# 3. Recent Progress

We summarize recent progress in AI for formal mathematical reasoning. A more extensive discussion can be found in the extended version of this paper (Yang et al., 2024b).

**Autoformalization.** Early attempts of machine learning for autoformalization were hampered by the lack of aligned informal-formal corpora (Kaliszyk et al., 2017; Wang et al., 2018; 2020). The emergence of in-context learning in LLMs opens up a new paradigm, requiring only a few expert-constructed demonstrations (Wu et al., 2022). In addition, *in*formalization is generally easier than formalization, and we can use LLMs to perform back-translation, i.e., generating synthetic aligned corpora by auto-*in*formalizing existing formal statements (Jiang et al., 2024; Azerbayev et al., 2023). Finetuning a smaller model on this synthetic data led to notable improvements in autoformalization.

As the bridge between informal and formal mathematics, autoformalization has three immediate applications: (1) data augmentation for neural theorem provers (Wu et al., 2022; Xin et al., 2024; Wang et al., 2025; Ren et al., 2025), (2) guiding theorem proving via informal proofs (Jiang et al., 2023), and (3) validating informal reasoning (Zhou et al., 2024a; Olausson et al., 2023).

**Neural Theorem Proving.** Deep learning has been widely used for learning heuristics to find proofs in formal systems (Vaezipoor et al., 2021; Lederman et al., 2020). Holophrasm (Whalen, 2016) was the first system to demonstrate the feasibility of training deep neural networks to guide proof search. This paradigm was expanded in GPT-f (Polu & Sutskever, 2020), which trained a single Transformer to generate proof steps. It enjoyed substantial gains from math pretraining (Sec. 2.1). Subsequent approaches have trained richer architectures and exploited zero-shot prompting of LLMs. We highlight several prominent ideas:

- *Expert iteration* alternates between (1) using the prover to attempt unsolved problems and, (2) if new proofs are found, using them as training data to improve the prover. It leads to a performance gain that diminishes after a few iterations (Polu et al., 2023; Lample et al., 2022).

- *Learning from mistakes*: Formal proof environments can provide error messages when a proof step fails. CO-PRA (Thakur et al., 2024) included error messages in the prompts, utilizing LLMs' in-context learning capability to reduce the odds of repeating similar mistakes.

- *Informal proof sketches*: Formal theorem proving has also benefited from informal proofs. Draft, Sketch and Prove (DSP) (Jiang et al., 2023) used LLMs to generate a "proof sketch" in natural language, and then autoformalized it in Isabelle. Lean-STaR (Lin et al., 2024) interleaved formal and informal reasoning steps in theorem proving in Lean.

- *Premise selection* involves retrieving useful lemmas in proving a theorem (Kühlwein et al., 2012; Irving et al., 2016; Mikuła et al., 2024). ReProver (Yang et al., 2023) applied retrieval for neural theorem proving, where it first retrieved lemmas from a mathematical library. COPRA also used retrieved lemmas as part of their LLM prompts.

**Verified Reasoning in Natural Language.** Reasoning problems expressed in natural language may be difficult to completely formalize. In such cases, we still want *some* form of verification. Several works have used neural networks as verifiers (Lightman et al., 2024; Yang et al., 2022; Ling et al., 2024). While neural verifiers cannot formally guarantee the validity of the reasoning, they nonetheless provided a boost in overall performance and faithfulness.

Alternatively, one can combine LLM-based autoformalization with formal problem solving. SatLM (Ye et al., 2023) and LINC (Olausson et al., 2023) converted the entire problem into appropriate formats and called SAT/SMT solvers to produce solutions. LogicGuide (Poesia et al., 2024b) used a formal system to constrain the step-by-step deductions from the LLM, producing chain-of-thought reasoning that alternates between formal reasoning and natural language.

**Formal Verification and Verifiable Generation.** AI can automate many tedious aspects in theorem proving for system verification, e.g., generating initial proofs (Sanchez-Stern et al., 2020) and refining existing proofs (First et al., 2023). Moreover, it is useful in SMT-based verification tasks, including inferring loop invariants (Si et al., 2020) and generating helper assertions (Mugnier et al., 2024). Integrating AI and verification has been explored in the formal method community. For example, Seshia (2015) proposed SID (Structure, Induction, and Deduction), a general framework that combines inductive and deductive reasoning for verification. In SID, machine learning models can serve as the inductive engine to generate artifacts (e.g., loop invariants or proofs), while formal systems such as Lean serve as the deductive engine to answer queries about these artifacts.

A closely related challenge is to simultaneously generate code and formal proofs. LLM-generated code can be buggy and insecure (Pearce et al., 2022; Perry et al., 2023). Coupling generation with formal verification is a natural way to prevent such failures. One possibility is to first develop a formally verified program in Coq or Lean, with AI assistance, and then translate it into a more efficient implementation using compilers. This approach establishes a direct arc between theorem proving and generation. Another possibility is to incorporate LLM-based code and proof generation into a high-level verification-friendly language like Dafny (Misu et al., 2024) and Verus (Lattuada et al., 2023).

## 4. Open Challenges and Future Directions

Formal mathematical reasoning presents a wealth of challenging problems for AI. Here, we explore several open challenges and promising directions.

### 4.1. Data

Scaling the training data in formal mathematics is hampered by the scarcity of human-created formal proofs. The Proof Pile dataset (Azerbayev et al., 2024), which aggregated proofs from six different formal languages, collected only 500MB of formal proofs—orders of magnitude smaller than its informal counterparts. The issue is more pronounced in research mathematics, where even informal data is limited.

Researchers are exploring different strategies to overcome data scarcity. The first is autoformalization. We have a substantial amount of informal math data. Since formal proofs can be verified easily, if a system successfully formalizes even a small subset of the informal math data, it can self-improve through expert iteration, potentially covering an increasingly larger set with each iteration (Sec. 3). Another approach is synthetic data generation. For example, the training data in AlphaGeometry and TongGeometry consisted solely of synthetic geometry problems and solutions (Trinh et al., 2024; Zhang et al., 2024a). Mathematical axioms entail all provable facts, in principle containing *infinite* data. By generating synthetic data, AI can potentially explore and learn from the vast space of possible mathematics, at a scale that can drastically surpass the pace of human-created training data. If the method can be generalized, it would help in completely new mathematical domains, where even informal data might be scarce.

Autoformalization and synthetic data were combined in AlphaProof (Google DeepMind, 2024), which autoformalized one million IMO-like informal problems into one hundred million formal theorems, whose proofs were synthetically

generated using expert iteration. It remains an open question to generalize this approach beyond domains where a large number of human-written problems are available, e.g., research mathematics. Those domains will likely require *conjecturing* new unseen statements (Poesia et al., 2024a).

Another promising strategy is knowledge transfer from different modalities. Specifically, code is closely related to mathematics, as both require symbolic reasoning. This similarity has been exploited to improve AI's general mathematical capabilities (Gao et al., 2023; Guo et al., 2024; Dubey et al., 2024), though it is still an open question how to leverage data-rich programming languages such as Python to enhance reasoning in *formal* mathematical languages.

## 4.2. Algorithms

**Autoformalization at Scale.** A major bottleneck in autoformalization is evaluating whether the autoformalized statement is logically equivalent to the ground truth. Proxy metrics such as BLEU (Papineni et al., 2002) do not correlate well with human judgment (Jiang et al., 2024), but relying on humans is not scalable. Possible ideas for better automated metrics include checking logical equivalence via automated provers (Murphy et al., 2024; Li et al., 2024c).

Autoformalization goes beyond translation, as some problems (e.g., IMO 2024 P5) may require complex reasoning, retrieving existing definitions, or even inventing new ones. For such problems, it is natural to break down the reasoning process into smaller steps, e.g., retrieving definitions before formalizing the statement or generating high-level sketches before formalizing the proof. We anticipate benefits from smaller steps and process supervision (Lightman et al., 2024; Lu et al., 2024) and call for autoformalization to be more interactive (Szegedy, 2020).

**Proof Search and Test-Time Compute.** Search is fundamental in many reasoning systems. Many neural theorem provers combine tactic generation with proof search (Fig. 5). The search strategy ranges from independent sampling of multiple candidates to sophisticated algorithms like Monte Carlo Tree Search (Lample et al., 2022). Scaling search to exploit vast test-time compute has emerged as a promising approach for both formal and informal reasoning (Google DeepMind, 2024; Zhang et al., 2024b; Xie et al., 2024b).

Many myths and trade-offs surrounding proof search remain unexplored. Is proof search really necessary, given that generating complete proofs can achieve lower latency (First et al., 2023; Xin et al., 2024). For a fixed compute budget, should we use smaller models with more search steps, or larger models (Wu et al., 2024a; Blaauwbroek et al., 2024)? How do different search algorithms compare? To answer these questions and guide the development of future provers, we need a systematic evaluation of existing methods. Such

an evaluation is lacking due to the inherent challenge in evaluating theorem proving in a fair and unified manner. It is unclear how to compare provers targeting different proof assistants. Even within the same proof assistant, a prover's performance is multifaceted and depends on resource constraints (e.g., hardware and time limits), making it difficult to consolidate performance into a single metric. A comprehensive evaluation that addresses these complexities would be immensely valuable. Despite these challenges, researchers are exploring various directions to improve proof search, such as developing value models to assess the promise of proof goals (Lample et al., 2022; Wu et al., 2024b).

Proof search alone does not solve theorem proving, where a fundamental challenge is a discrete, infinite action space. Proof search cannot succeed if the model cannot produce high-quality actions in the first place. In the context of theorem proving, mathematical creativity can manifest as actions exceeding the current model's capabilities, akin to the "divine move (神之一手)"—a legendary concept in Go. We would not expect to find them if the action space were an infinite, unstructured list. Fortunately, mathematics is structured, making it possible—though still challenging—to find the divine moves (Gowers, 2022). Next, we discuss several ways of leveraging structures in mathematics.

**Exploiting Hierarchies and Learning Abstractions.** Theorems are built upon smaller lemmas, which in turn break down into even simpler subgoals. Several existing theorem provers leverage this hierarchical structure. Draft, Sketch, and Prove (Jiang et al., 2023) transformed informal proofs into formal "proof sketches"—skeletons of formal proofs with "holes", i.e., open goals left unproven, yielding a hierarchical structure. POETRY (Wang et al., 2024) recursively decomposed goals in proof sketches using an LLM. While these works demonstrated the potential of hierarchical decomposition, it is still a significant challenge to decompose realistic high-level goals with current LLMs.

Abstraction is central to human mathematical practice. We first learn natural numbers through counting; years later, those operations show up in solving equations but do not require as much attention anymore. In interactive theorem proving, abstractions can be encapsulated in new definitions, lemmas, and tactics. While most existing methods assume they are predefined and fixed, recent work has explored learning abstractions. For instance, LEGO-Prover (Xin et al., 2023) used LLMs to propose and prove new lemmas, integrating them into its library to help prove further theorems. Lemma mining from existing proof corpora has also been explored (Kaliszyk & Urban, 2013; Zhou et al., 2024b). These lemmas, not explicitly factored out by humans, are still useful for automation. On learning *tactics*, Peano (Poesia & Goodman, 2023) and LEMMA (Li et al., 2022) have proposed to learn simple proof strategies from

an agent's own solutions to past problems, in a bootstrapping fashion. However, these approaches have so far been limited to simpler formal systems, and it is still an open challenge to synthesize entirely new tactics in full-fledged formal theorem proving languages like Lean.

**Incorporating External Knowledge.** Formal mathematical reasoning can benefit from explicitly retrieving and incorporating knowledge from databases of existing lemmas/definitions. ReProver (Yang et al., 2023) and CO-PRA (Thakur et al., 2024) demonstrated performance gains through standard retrievers like BM25 (Robertson et al., 2009) and Dense Passage Retrieval (Karpukhin et al., 2020). A promising direction involves developing retrieval methods tailored to formal math, e.g., structured or neurosymbolic retrievers. Another avenue is to grow the knowledge base dynamically. For example, the system could decompose high-level proof goals into subgoals, cache a subset of these subgoals as modules, and use them in subsequent proof efforts. Deciding which subgoals are "interesting" enough to be modularized in this way is a challenge.

### 4.3. Formal Verification and Verifiable Generation

Like AI4Math, we envision a growing need for formal reasoning in AI-based software and hardware generation, with assurance of correctness and security. While syntactical correctness can be guaranteed by constrained decoding (Beurer-Kellner et al., 2024), ensuring semantic properties, such as those validated by compilers, remains a challenge. Moreover, formal reasoning can help programmers understand AI-generated code (Ferdowsi et al., 2024).

Formal verification poses unique challenges. For example, a necessary but challenging step is encoding the target system and the correctness requirements in the proof assistant, similar to formalizing mathematics. However, while mathematical statements tend to assert properties of established mathematical objects, statements in formal verification typically concern bespoke procedures and datatypes. Also, proofs tend to be more repetitive and heavy on case-splits and inductive reasoning about recursive functions and datatypes. Finally, unlike statements in mathematics research, real-world software and hardware systems are characterized by large codebases and frequent changes. For instance, seL4 (Klein et al., 2009) consists of about 200,000 lines of specifications and proofs in Isabelle. Verifying these systems requires not only theorem proving but also rigorous management of specifications and proofs—an exciting yet underexplored direction for AI (Ringer et al., 2019).

It is natural to couple formal verification and AI-based generation to simultaneously generate code, formal specifications (i.e., pre/post-conditions, loop invariants, and helper assertions), and proofs. Then a program verifier or a theorem prover can check if the code is consistent with the specifications and proofs. This approach has been explored in recent research (Sun et al., 2024a; Yang et al., 2024a) and can potentially reduce verification efforts and enhance software and hardware reliability. However, a key challenge is to ensure the trustworthiness of the generated specifications—that they accurately reflect developers' intent.

## 5. Milestones and Success Measures

Inspired by the levels of automation for self-driving cars (SAE, 2024), we propose levels for AI-based formal mathematical reasoning. A more extensive discussion can be found in the extended version of our paper (Yang et al., 2024b).

### 5.1. Autoformalization

- *Level 0, representing knowledge in formal systems to support manual formalization*: Achieved by modern proof assistants such as Lean.

- *Level 1, generating autoformalization candidates and collecting human feedback*: LLMs can often generate good autoformalization candidates, but we need a system to gather and store human feedback, e.g., bug fixes and revisions made by humans to make the candidates usable. The system would also keep informal-formal pairs synchronized as the formal statements continue to develop.

- *Level 2, robust and faithful translations between informal and formal*: Model performance at this level could be assessed using human-curated benchmarks, including challenges from the ICML 2024 Math-AI workshop (mat, 2024; Huang et al., 2024b), ProofNet (Azerbayev et al., 2023), Herald (Gao et al., 2025), and Con-NF (Liu et al., 2025). However, a major obstacle is how to automatically evaluate autoformalized statements (Sec. 4.2).

- *Level 3, inferring missing information and flagging situations when a gap cannot be filled*: Implicit assumptions and hand-waived proof steps frequently pose challenges in formalizing mathematics. Bridging these information gaps requires robust reasoning capabilities: Proof gaps may be filled by neural or symbolic theorem provers, while missing assumptions can be resolved using abductive reasoning or counterexamples (Bundy et al., 2005; Blanchette & Nipkow, 2010). The main challenge is for the models to identify gaps—such as assessing the likelihood that a statement is provable or can be adjusted—even when it cannot bridge the gap immediately.

- *Level 4, self-correcting erroneous or inconsistent inputs by understanding human intentions*: At this stage, the autoformalization model focuses more on capturing human intentions and may rely on its own self-consistency to eliminate errors. Advancements here will be closely

linked to natural language reasoning (Sec. 5.3).

- *Level 5, proposing novel definitions that can reduce proof complexity*: AI can serve as "theory builders" that reshape the proving process through better abstraction or concept formulation. For instance, filters (i.e., a set of sets satisfying certain properties) are rarely taught in standard math curricula but have become convenient for formalizing limits in various proof assistants. Automatically devising definitions like filters is what we hope AI can achieve.

## 5.2. Theorem Proving

- *Level 0, checking formal proofs*: Achieved by modern proof assistants such as Lean.

- *Level 1, assisting humans to develop formal proofs by suggesting definitions, lemmas, proof steps, etc.*: Library search engines like LeanSearch (Gao et al., 2024) and proof completion "copilots" (Dohmke, 2023; Song et al., 2025) can be highly helpful, though humans are still responsible for the main job of developing the proof.

- *Level 2, human-implemented tactics for proof automation*: These are domain-specific procedures for automating certain classes of proofs, e.g., `omega` and `nlinarith` can automatically solve many equalities and inequalities. This level involves no machine learning but mostly human-engineered domain-specific methods to produce proofs.

- *Level 3, proving simple theorems automatically in a domain-general fashion*: Recent neural theorem provers are domain-agnostic and evaluated on benchmarks targeting this level, e.g., CoqGym (Yang & Deng, 2019), LeanDojo (Yang et al., 2023), MiniF2F (Zheng et al., 2022), and PutnamBench (Tsoukalas et al., 2024). These systems are limited to relatively simple proofs, typically not the most time-consuming ones, but they can still be useful for closing simple gaps in larger proofs.

- *Level 4, contributing to formalization projects autonomously*: Beyond proving individual theorems, AI should be able to break down larger results, state new definitions and lemmas, and potentially explore different alternatives as the project develops. Evaluation may require new benchmarks constructed from GitHub metadata, such as issues and commits, of real-world formalizations.

- *Level 5, solving problems and discovering new math beyond the human level*: Out of reach for current AI systems. One challenge will be to measure progress meaningfully towards this open-ended goal, since our current evaluations only test knowledge of existing mathematics.

## 5.3. Verified Reasoning in Natural Language

- *Level 0, stepwise natural language reasoning w/o verification*: Chain-of-thought (Wei et al., 2022) is effective but

its reasoning can be brittle, incorrect, or unfaithful (Shi et al., 2023; Lanham et al., 2023; Ling et al., 2024).

- *Level 1, stepwise natural language reasoning with neural verification*: Neural verifiers can improve reasoning, e.g., by selecting the best output from many generated candidates (Cobbe et al., 2021). Evaluation can be done using standard benchmarks like MATH (Hendrycks et al., 2021) or by directly measuring whether the model can identify reasoning errors (Hong et al., 2024; Zheng et al., 2024a). Standard benchmarks like MATH suffer from data contamination (Dong et al., 2024). To mitigate the issue, dynamically generated benchmarks such as GSM-Symbolic (Mirzadeh et al., 2024) or private benchmarks such as FrontierMath (Glazer et al., 2024) may be useful.

- *Level 2, tool-integrated reasoning using SymPy, NumPy, etc.*: Models can leverage external tools to perform computation that neural networks struggle to learn reliably, e.g., numerical calculations and symbol manipulation. They can be evaluated on math problems requiring intricate computations, e.g., the MATH dataset or AIME.

- *Level 3, Reasoning seamlessly in natural language and formal systems such as Lean*: Instead of formalizing the entire problem (Zhou et al., 2024a), the model can interleave natural language with formal reasoning, selectively determining which parts of reasoning to process using formal systems. Benchmarks should evaluate not only the final answer but also the quality of reasoning, and they should include math problems that resist complete formalization, e.g., IMO as taken by human contestants, without manual formalization as AlphaProof did.

- *Level 4, complex mathematical reasoning and planning in real-world applications*: Applications of complex reasoning often contain mathematical components along with other components such as commonsense and human preferences. In scenarios like travel planning (Xie et al., 2024a) or calendar scheduling (Zheng et al., 2024b), AI could formulate the task as a constraint satisfaction problem and solve it using appropriate solvers. Achieving this capability would enable a wide range of applications.

## 5.4. Formal Verification and Verifiable Generation

- *Level 0, code generation without verification*.

- *Level 1, verifying and synthesizing small programs with simple properties*: Several benchmarks have targeted *verification* at this level (Loughridge et al., 2024; Lohn & Welleck, 2024; Aggarwal et al., 2024), but current models still fall short. We are not aware of large-scale benchmarks for synthesizing verified code together with a formal specification, even at Level 1 (Brandfonbrener et al. (2024), Ye et al. (2025), and (Thakur et al., 2025) introduced small-scale benchmarks in this direction).

- *Level 2, verifying and synthesizing entire projects with complex functional and security properties*: This level requires decomposing large systems into smaller verifiable components, a task currently performed by humans (Gu et al., 2016) but may be tackled by advanced AI agents capable of planning and problem-solving to navigate the intricate dependencies and interactions in large codebases. Benchmarks should incorporate project-level context, e.g., extracted from existing verification projects systems (Leroy et al., 2016; Zhang et al., 2024c).

- *Level 3, proof and system maintenance*: System designs and implementations constantly evolve, and so must their proofs to stay synced. AI should provide assistance when developers update the system or refactor proofs (Ringer, 2021). To evaluate this capability level, benchmarks can be constructed from the change history of verified systems (Reichel et al., 2023). These benchmarks should capture a variety of scenarios, including minor bug fixes, major feature additions, and comprehensive refactoring.

- *Level 4, helping users generate, explain, and debug formal specifications*: AI should aid users in writing specifications, which requires abstracting and converting user requirements into formal languages. The evaluation can leverage verified codebases. Instead of generating proofs and code given the specifications, we treat them as ground truth and use them to evaluate specifications generated by the model. The system should be interactive to engage with users, offering suggestions and clarifications.

## 6. Alternative Views

We argue that formal mathematical reasoning is essential to address data scarcity and lack of verifiability in the informal approach (Sec. 2.1). **An alternative view is that these challenges could be addressed by neural networks alone, without formal systems.** For example, OpenAI o1 and DeepSeek-R1 have demonstrated impressive capabilities on problems with numeric answers. However, it remains unclear if they can solve problems requiring rigorous intermediate steps, such as theorem proving. Neural verifiers could add rigor to the informal approach, but their success hinges on whether neural networks can achieve (1) self-verification/correction and (2) easy-to-hard generalization (Sun et al., 2024b). Despite extensive studies on self-verification, current LLMs struggle with *intrinsic self-verification*—verifying their own generations without external feedback (Huang et al., 2024a; Stechly et al., 2024; Gou et al., 2024a; Zheng et al., 2024a; Gu et al., 2024).

## 7. Conclusion

We advocated for formal mathematical reasoning as an important complement to the informal approach, highlighting its potential to advance AI in math and verified system de-

sign. We hope to present coherent perspectives that unite previously fragmented efforts across different fields, fostering discussion, community building, and a future roadmap.

## Acknowledgements

We gratefully acknowledge Jeremy Avigad, Albert Q. Jiang, Zhaoyu Li, Peter O'Hearn, Daniel Selsam, Sanjit A. Seshia, Armando Solar-Lezama, and Terence Tao for providing valuable feedback on an initial version of this paper.

## Impact Statement

This position paper aims to advance AI for mathematical reasoning and its application in verifiable software and hardware system design. There are many potential societal consequences of these areas, none of which we feel must be specifically highlighted here.

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
