# OpenReview forum: "Position: Formal Mathematical Reasoning—A New Frontier in AI"
_ICML.cc/2025/Position_Paper_Track — ICML 2025 Position Paper Track spotlightposter_

### Official Review · Reviewer_mAgc · 2025-03-12

**Significance:** 4
**Argument Clarity:** 4
**Rating:** 5
**Confidence:** 4

**Questions:**

I have no major questions.

If anything there could be some discussion as to what areas of mathematics are more ripe for autoformalization rather than others. For example, is research into PDEs inherent harder to formalize than algebraic combinatorics? It seems so.

**Discussion Potential:**

3

**Paper Summary:**

The paper argues for research into formal mathematical reasoning, this is defined as mathematical reasoning grounded in formal systems, like Lean or Homotopy (or dependent) Type Theory.

The call to action is that "AI-based formal mathematical reasoning has reached an inflection point, with significant progress in the
near future" possible. The authors posit that if LLMs (or other generative systems) can auto-formalize informal, human generated mathematics, then progress in mathematical research (and other fields, including applied mathematics, etc.) can be progress faster, eventually generating proofs of theorems or even conjectural theorem statements. Eventual verified reasoning in natural language is another target goal. The authors argue that this can be achieved essentially through a 3-step process:

1. Autoformalization of natural language based reasoning, i.e. the turning of informal problem statements into formal problem statements.

2. Neural Theorem Proving: the process of reasoning based on the formalization. Typically this involves turning the goal statement into a hierarchy of subgoals and then suggestions of possible tactics for proving each subgoal. This is essentially a proof tree that can be searched using clever methods.

3. Deformalization: Turning the proof or major steps of the proof into informal human language.

Recent progress both in formalization, e.g. with Lean, and automated mathematical reasoning engines, e.g. AlphaProof, are reviewed.

Challenges, including scarcity of good data for advanced mathematics and its corresponding formalization, lack of good algorithms for searching the space of tactics for proving statements, and automated abstraction, are also discussed.

Finally the paper concludes with a conjectural hierarchy of capabilities, usually listed from level 0 (= current SOTA) to level 5 (= almost sci-fi capabilities) for four capabilities: auto formalization, theorem proving, verified reasoning in natural language, and formal verification and verifiable generation.

The main opposing position discussed is that neural networks alone, without using formal systems such as Coq or Lean can address the challenges discussed.

## Update After Rebuttal

It is a strong paper and I recommend acceptance. I thank the authors for considering whether different areas of math are more amenable to formalization.

**Position:**

Yes

**Position In Title:**

Yes

**Related Work:**

4

**Strengths And Weaknesses:**

I would enumerate three main strengths of the paper:

I. Discussion of current formalization systems such as Lean make a compelling case for how human-AI collaborative reasoning can proceed. Current SOTA for automatic decomposition of a goal statements into sub-goals and suggested tactics for proving sub-goals seems ripe for improvement with LLMs finely-tuned on math data sets. Lean's mathlib is currently quite small as a data set, but if AI assistants make formalization easier, then it stands to reason that curated/verified data sets such as mathlib seem likely to experience exponential growth.

II. The line of research carves out an implied role for human mathematicians: to serve as creative, conjecture-making machines. It now seems that human intuition for making conjectures will become the main prize of research mathematics, and technical machismo will become secondary. Dreamy far-off conjectures could provide long distance landmarks for autoformalization and neural theorem proving. Additionally, the ability to then recurse on these AI+human mathematician collaborations seems be a major force multiplier in making discoveries in modern mathematics, and hence science and engineering at large.

III. The Level system for measuring the four capabilities surveyed in the paper seems well though out and I can see widespread acceptance in the community for them. Level 4 in Section 5.3 seemed especially desirable.

I cannot identify any major weaknesses.

**Support:**

4

---

> ### Author Rebuttal · Authors · 2025-04-01
>
> Dear reviewer,
>
> Thank you for taking the time to review our paper and provide feedback! Below we address your comments and concerns. Please feel free to follow up if you have further questions.
>
> ## What areas of mathematics are more amenable to formalization?
>
> In principle, Lean can be used to formalize nearly all of mathematics, as it encompasses first-order logic and can express ZFC set theory. However, some areas may require more effort to formalize than others (for both humans and AI). For example, formalizing analysis (including PDEs) tends to be more challenging than algebra. This is partly due to the fact that mathlib is more mature in algebraic, and many early Lean users are mathematicians working on algebra, e.g., Kevin Buzzard. Additionally, analysis often involves lengthy and intricate computations better suited to computer algebra systems (CAS) like SymPy or Mathematica. At present, Lean lacks a mature CAS, and while building one is feasible, it would require substantial time and engineering.

---

> > ### Comment · Reviewer_mAgc · 2025-04-05
> >
> > thank you for the comment, i look forward to reading the revision.

---

### Official Review · Reviewer_jVgm · 2025-03-12

**Significance:** 4
**Argument Clarity:** 4
**Rating:** 5
**Confidence:** 4

**Questions:**

Questions:
- Could post-training techniques like logic-guided fine-tuning further enhance pre-trained LLMs’ formal reasoning capabilities, and if so, what steps might researchers take to explore this direction?
- Could alternative architectures (e.g., graph or neuro-symbolic models) better address hierarchical or non-Euclidean mathematical structures compared to standard LLMs?
- How might the field balance mathematical rigor with accessibility—e.g., through user-friendly tools—to realize AI4Math’s full potential beyond research communities, and is democratizing entry barriers a worthwhile pursuit despite potential trade-offs in technical precision?

**Discussion Potential:**

3

**Paper Summary:**

Summary: This paper illustrates current advances, open challenges, and future directions in AI-driven formal mathematical reasoning, emphasizing its dual impetus for advancing theorem proving and verified system design while advocating concerted research efforts toward AI in math and verified system design.

## update after rebuttal
Thank the authors' detailed responses, which have solved my questions and concerns. I have no more question and would suggest an acceptance.

**Position:**

Yes

**Position In Title:**

Yes

**Related Work:**

4

**Strengths And Weaknesses:**

Strengths:
- Formal Mathematical Reasoning is a promising and important direction, but current research largely focuses on achieving higher accuracy using informal methods, neglecting the significance of formal methods. This makes the appearance of this paper meaningful.
- The paper is generally well written and structured, providing a detailed and progressive elaboration that systematically progresses from current methodologies to open challenges and potential research directions.
- The challenges outlined in Section 4 capture some of the core issues of the field, with data scarcity and verifiability being critical problems for the advancement of AI in mathematics, which enhances the persuasiveness of the main position.
- 'Levels for AI-based formal mathematical reasoning' in Section 5 is well-structured and accompanied by widely recognized examples, presenting a clear and detailed set of criteria that holds long-term significance and provides clear guidance for future research.

Weaknesses:
- The paper highlights data and algorithms; however, post-training strategies (e.g., logic-guided fine-tuning) might enhance the formal reasoning ability of the model, but this has not been deeply studied.
- While the paper focuses on LLMs, alternative architectures (e.g., graph or neuro-symbolic models) are not widely discussed, despite their potential to offer better adaptability to hierarchical or non-Euclidean mathematical domains.
- While Lean is a widely used and highly regarded formal proof assistant, it has limitations, such as high entry barriers and potential challenges in addressing complex or advanced mathematical problems. The paper, however, does not thoroughly address the development of novel methods or the broader adoption of AI4Math.
- The arguments about "data scarcity" rely solely on problems in competitions like IMO without analyzing some real-world distributions in advanced fields like algebraic geometry, indicating a narrow empirical scope.

**Support:**

4

---

> ### Author Rebuttal · Authors · 2025-04-01
>
> Dear reviewer,
>
> Thank you for taking the time to review our paper and provide feedback! Below we address your comments and concerns. Please feel free to follow up if you have further questions.
>
> ## Post-training techniques such as logic-guided finetuning
>
> We agree that post-training plays a key role in enhancing LLMs' reasoning abilities. Our paper touches upon several topics in post-training, including RL from proof assistant feedback (Sec. 3) and knowledge transfer from different proof languages and modalities (Sec. 4.1). Regarding “logic-guided finetuning”, we’re not sure what it refers to and would appreciate it if the reviewer could clarify the term or point us to relevant references.
>
> ## Alternative architectures beyond Transformers
>
> We agree with the reviewer on the importance of exploring alternative model architectures for mathematical reasoning. Transformers as in LLMs are particularly scalable for large-scale pretraining, making them suitable for building general-purpose AI mathematical models that need to absorb a broad range of mathematical knowledge. Alternatives—such as graph neural networks and neuro-symbolic architectures—may be well-suited for certain mathematical domains due to their inductive biases. However, integrating them with large-scale learning remains an open challenge if the goal is to build general-purpose AI mathematicians. That said, they have been proved effective for specific mathematical domains e.g., graph neural networks are widely used to model combinatorial optimization problems [O].
>
> * [O] Cappart, Quentin, et al. "Combinatorial optimization and reasoning with graph neural networks." Journal of Machine Learning Research 24.130 (2023): 1-61.
>
>
> ## How to make formal reasoning more accessible?
>
> We agree that accessibility is critical. Our paper discusses integrating informal and formal reasoning (Sec. 3 and 5.3) as one possible approach to strike a balance between accessibility and mathematical rigor. We also appreciate your idea on building user-friendly tools and will expand on this in future revisions. Thank you for the helpful feedback!

---

### Official Review · Reviewer_zg3j · 2025-03-13

**Significance:** 3
**Argument Clarity:** 2
**Rating:** 4
**Confidence:** 5

**Questions:**

Please see weaknesses.

**Discussion Potential:**

4

**Paper Summary:**

This paper talks about the topic of AI for formal mathematics. In its own words, the paper “advocates formal mathematical reasoning as an indispensable component in future AI for math, formal verification, and verifiable generation.” It reviews the literature, and claims to define milestones to measure future success.

**Position:**

Yes

**Position In Title:**

Yes

**Related Work:**

3

**Strengths And Weaknesses:**

### Strengths:

The topic is important and timely. It would be very good to have a position paper on the topic of formal mathematical reasoning at ICML.

Literature review is broad.

Writing is clear, but it definitely needs proofreading.

-------------------


### Weaknesses:

The position of the paper remains mostly vague. Reading the abstract and then conclusion, I find it rather hard to see anything interesting or new in them. Most researchers with some minimal familiarity with the topic of AI4Math would probably find nothing new, interesting, or even controversial in most of the paper. I think the writing of the abstract and conclusion, and some other parts of the paper are weak. I suggest authors completely revise them.

One instance where the position of the paper becomes clear is the Alternative Views section, whether one agrees with it or not. What is presented in the alternative views, at the very end of the paper, can appear much earlier to clarify the position of the paper. A reader does not read the paper backwards.

ICML Position paper track indicates: “The goal of this track is to highlight papers that stimulate (productive, civil) discussion on timely topics that need our community’s input.  Controversial topics are welcome.” The topic of this paper is timely, but the paper needs to elaborate further on the positions that are out there, and the specific position that it wants to take.

The literature review, at times, appears rosy to me. It is good to sing the praises of the prior work in the literature, but when the paper claims “AlphaProof leveraged formal reasoning to become the first AI to achieve the silver medal level in IMO.”, I find that quite unfounded by the standards of scientific research. How long did it take for AlphaProof to prove the problems of IMO 2024? Was it the same as human participants of IMO? I understand why DeepMind might want to put out a news article claiming the silver medal, but I’m not sure if this news article should be cited in ICML proceedings. Was AlphaProof evaluated on the benchmarks of the community like miniF2F, or on the previous IMO problems. Is it clear to the authors how this AI model works and is there enough knowledge about it for anyone to reproduce it? I personally am very curious to know how this AI model performs in proving the problems in IMO 1985, and not only IMO 2024. I also would like to know the accuracy of this so-called silver medalist model on miniF2F valid and test set. But, I can’t find the answer to any of these questions. So, I find such claims in the paper problematic.



I was expecting the paper to talk about training set contamination at least once. This is an important topic for both formal mathematics and informal one. Are there any models that are upfront about their training set? Leandojo is a very good example that deserves a mention for its effort for avoiding training set contamination. But, are there any other models? What is the position of the paper in this regard? Is it important for the community to adopt better evaluation practices when it comes to formal and informal mathematical reasoning with AI models?


I think it would have been nice for the paper to give credit to the tools developed by mathematicians, for example the symbolic engine used by AlphaGeometry, or other tools in Lean such as nlinarith that LLMs leverage quite frequently. Most of what LLMs can prove from miniF2F heavily relies on those solvers, and it would be proper for a position paper to give credit to them.


I might have missed it, but the paper does not discuss the importance of mathematical reasoning in informal language, and how it can help the reasoning in formal language. For example, when one asks an advanced LLM to write a formal proof in Lean, does it matter whether that model has some knowledge about the topic in informal language. Is there any evidence that purely training an AI model on formal language data can lead to good generalization?


Highlighting the weaknesses and strengths of existing models in formal and informal mathematics, by giving some examples can make the paper more grounded in reality. For example, showing a difficult problem from miniF2F that an AI model has been able to prove, how long the proof is, what is the subject, etc. Similarly, are there trivial problems from miniF2F that best models cannot prove in formal language? Since the paper wants to do community building and attract others to enter the field of formal reasoning, it would make sense to give a clearer view of the capabilities of current models.


Paper makes some claims about the easiness of pre-college math and the topics that LLMs can learn using informal language. And then it claims that for advanced mathematics, data is scarce, so we need formal verification. However, I find these claims not quite accurate and perhaps not in line with some of the other claims in the paper. Have LLMs trained on informal math mastered the topics in pre-college math? They have not, and the best models still make trivial mistakes on topics that data is not scarce in informal language. At the same time, even in pre-college math, formal data is scarce. So, those arguments in the paper don’t seem educated enough to me.

**Support:**

2

---

> ### Author Rebuttal · Authors · 2025-04-01
>
> Dear reviewer,
>
> Thank you for taking the time to review our paper and provide feedback! Below we address your comments and concerns. Please feel free to follow up if you have further questions.
>
> ## Clarifying the position
>
> Our central position is that **AI4Math systems based on LLMs alone face challenges in data scarcity and verifiability, and these challenges are best addressed by integrating informal reasoning with formal methods grounded in systems like Lean.** While this position may be familiar to communities centered around Lean and formal mathematics (related positions discussed in our reply to reviewer 9EnC), it is far from a consensus in the machine learning community. State-of-the-art LLMs (e.g., GPT-4o, Claude 3.7 Sonnet) do not incorporate formal mathematics in any substantial way. Therefore, we believe it is valuable to discuss and advocate for formal mathematical reasoning within the ICML community.
>
> Thank you for your suggestions on the writing. We’ll incorporate them in the next revision, e.g., using alternative views to help clarify the position in the introduction
>
>
> ## Closed-source systems such as AlphaProof
>
> Thank you for the suggestions on addressing closed-source systems more clearly. For AlphaProof, the only public references we are aware of are [DeepMind’s blog](https://deepmind.google/discover/blog/ai-solves-imo-problems-at-silver-medal-level/) and Thomas Hubert’s talk [N]. We will update the paper to clarify this and provide the necessary context.
>
> Although AlphaProof is not open-source, we see it as an exciting milestone and are encouraged by the open efforts underway (e.g., by Numina) to replicate its success. The field has often seen rapid open-source progress following such releases (e.g., GPT-3 -> LLaMA, o1 -> R1).
>
> * [N] Thomas Hubert “[AlphaProof](https://www.youtube.com/live/3gaEMscOMAU)”
>
>
> ## Data Contamination
>
> We appreciate your suggestion. The discussion on data contamination is currently in Appendix C.2 (Lines 1347–1355): We acknowledge that data contamination is a known issue in standard mathematical reasoning benchmarks (e.g., MATH). Approaches such as using dynamically generated test sets or keeping the test set private can help mitigate this problem. We will move it into the main paper for greater visibility.
>
>
> ## Symbolic tools such as `nlinarith`
>
> Thank you for pointing this out. We discussed mathematician-implemented tactics like `omega` in Sec. 5.2 and will expand the discussion to include `nlinarith`.
>
>
> ## How informal mathematical reasoning helps formal reasoning
>
> We believe informal mathematical reasoning supports formal reasoning in two key ways: (1) LLM-based provers acquire mathematical reasoning skills through pretraining on informal texts [P], rather than relying solely on formal data; and (2) formal provers often interleave informal comments and tactics during proof synthesis (e.g., DeepSeek-Prover and Lean-STaR), or directly use informal proofs to guide formal theorem proving (e.g., Draft-Sketch-Prove).
>
> * [P] Azerbayev, Zhangir, et al. "Llemma: An open language model for mathematics." ICLR 2024.
>
>
>
> ## Examples highlighting the strengths and weaknesses of existing models
>
> Thank you for the suggestions, and we’ll add more examples. For instance, LLM-generated proofs on MiniF2F problems often rely heavily on automation tactics such as `linarith` and `norm_num`.
>
>
> ## LLMs making trivial mistakes on pre-college math
>
> Our paper acknowledged that LLMs still exhibit reasoning flaws (currently in Appendix C.2 Line 1350–1352 and will be mentioned in the main paper in the next revision) and chain-of-thought reasoning generated by LLMs can be brittle or incorrect (Sec. 5.3 Line 386–389). These flaws strengthen our point that formal mathematics is needed for robust and rigorous reasoning.
>
>
> ## Data scarcity in mathematics for advanced mathematics
>
> Our main argument is that informal corpora for advanced mathematics are significantly scarcer than those for pre-college math. However, automatic feedback from proof assistants (as discussed in Sec. 4.1) could help enhance models aimed at advanced mathematics, thereby mitigating the data scarcity issue.

---

> > ### Comment · Reviewer_zg3j · 2025-04-02
> >
> > I thank the authors for their clear response. The clarifications and adjustments that you describe will improve the paper in my view. I will increase my score and I think your paper can be a good contribution.
> >
> > Your answer about informal mathematical reasoning is good, though the rebuttal does not mention doing any adjustments about it in the paper. Given your explanation of the position of the paper in the rebuttal, I suggest describing the interaction between informal mathematical reasoning and formal reasoning in further detail in the paper, and perhaps at a specific section or subsection. The paper explains some of such interactions in various parts of the paper, and at one point, the paper suggests that formal and informal reasoning should complement each other. But, it appears to me that this discussion goes exactly to the heart of your position, and currently, it is dispersed here and there mostly at later parts of the paper.
> >
> > My suggestion is that the paper explicitly talks about how it envisions formal reasoning and informal reasoning should complement each other, discuss the evidence for and against their fruitful collaboration, and the limitations that the paper thinks will arise when one uses only formal reasoning or only informal reasoning. I can find some of these answers already in the paper, but as I mentioned, they are dispersed here and there. And the introduction mostly focuses on the limitations of informal mathematical reasoning.

---

> > > ### Author Response · Authors · 2025-04-04
> > >
> > > Thank you very much for updating your score and for the thoughtful feedback! We will revise the paper and especially the introduction to clearly articulate how informal and formal reasoning should work together to improve LLMs’ mathematical reasoning capabilities, incorporating our discussions in the rebuttal and consolidating the currently dispersed points throughout the paper.

---

### Official Review · Reviewer_9EnC · 2025-03-15

**Significance:** 1
**Argument Clarity:** 3
**Rating:** 2
**Confidence:** 3

**Questions:**

N/A

**Discussion Potential:**

1

**Paper Summary:**

The paper surveys progress in formal mathematical reasoning and outlines how the field might evolve in different directions. It provides new measures of success, as milestones that may be achieved in future and argues that formal mathematical reasoning will be an indispensable component in future AI for math.

**Position:**

Yes

**Position In Title:**

No

**Related Work:**

2

**Strengths And Weaknesses:**

The paper clearly explains some features of formal reasoning, is one of the first to provide a quick discussion of NuminaMath model, and contains a discussion. Nonetheless, the main issue is that this reads more like a survey that also introduces a taxonomy for future progress (the levels in section 5) - rather than an opinion. This is also evident from the abstract, when it is mentioned: " We summarize existing progress, discuss open challenges, and envision critical milestones to measure future success."

The opinion that formal mathematical reasoning will be an indispensable component in future AI for math is not really new. Leading mathematicians such as Terence Tao (and even more, Kevin Buzzard) have claimed this for a few years now.

This is compounded by some minor issues:
- Some inaccurate statements. E.g., "The informal approach faces challenges in dealing with advanced mathematics. First, high-quality training data is inherently scarce in advanced mathematics." and later "We argue that formal mathematical reasoning is essential
to address data scarcity and lack of verifiability in the informal approach".
It is actually the other way around: For informal mathematics, there is a lot of training data, some even high-quality (NuminaMath dataset, https://huggingface.co/datasets/AI-MO/NuminaMath-TIR). For formal mathematics, there are few dedicated datasets (miniF2F, ProofNet) other than the various formal libraries. So formal math actually suffers from training data (unless you argue that one can train directly on the formal libraries, but this has issues on its own, some of which were discussed in https://arxiv.org/abs/2412.15184)
- Missed references/not up to date (for Geometry, only AlphaGeometry is mentioned; Newclid is a direct successor to it, and TongGeometry also surpasses it)

All in all, I think this would make a good survey, but it is not really a position paper, since the position is already well-known.

**Support:**

4

---

> ### Author Rebuttal · Authors · 2025-04-01
>
> Dear reviewer,
>
> Thank you for taking the time to review our paper and provide feedback! Below we address your comments and concerns. Please feel free to follow up if you have further questions.
>
> ## Suitability for the position paper track
>
> We believe our paper aligns well with the [ICML 2025 Position Paper CFP](https://icml.cc/Conferences/2025/CallForPositionPapers). Our central position is that **AI4Math systems based on LLMs alone face challenges in data scarcity and verifiability, and these challenges are best addressed by integrating informal reasoning with formal methods grounded in systems like Lean.** We argue for the importance of this direction and outline open challenges, future directions, and milestones, which is consistent with `Position papers make an argument for a viewpoint or perspective about what should be done … Examples include (but are not limited to) an argument in favor of or against a particular research direction` in the CFP.
>
> The CFP also encourages referring to ICML 2024 position papers for guidance. Many of them (e.g., [A–E]) adopted a similar structure to advocate for a research direction. Several ([A, B, C]) used similar language in their abstract as our statement called out by the reviewer (`We summarize existing progress, discuss open challenges, and envision critical milestones to measure future success.`)
>
> * [A] Papamarkou et al. "Position: Bayesian Deep Learning is Needed in the Age of Large-Scale AI." ICML 2024
> * [B] Papamarkou et al. "Position: Topological Deep Learning is the New Frontier for Relational Learning." ICML 2024
> * [C] Spangher et al. "Position: Opportunities Exist for Machine Learning in Magnetic Fusion Energy." ICML. 2024.
> * [D] Morris et al. "Position: Future directions in the theory of graph machine learning." ICML 2024
> * [E] Lindauer, Marius, et al. "Position: A Call to Action for a Human-Centered AutoML Paradigm." ICML 2024
>
> ## Related positions by Mathematicians
>
> Thank you for pointing out the positions of Tao and Buzzard. We will revise the paper to discuss these positions and how they relate to ours.
>
> Our position is complementary to those of Tao and Buzzard (expressed in their recent blogs and talks [F–M]). They advocate for using proof assistants (as opposed to pencils and papers) to formalize mathematics, enabling rigorous proof verification and large-scale collaboration. While they touch on AI’s role in automating parts of this process, their primary focus is on how AI may assist mathematicians. In contrast, our position focuses on **how formal math can benefit AI** by improving its capabilities in mathematical reasoning, formal verification, and verifiable code generation.
>
> * [F] Terrence Tao “[The Potential for AI in Science and Mathematics](https://www.youtube.com/watch?v=_sTDSO74D8Q)”
> * [G] Terrence Tao “[A slightly longer Lean 4 proof tour](https://terrytao.wordpress.com/2023/12/05/a-slightly-longer-lean-4-proof-tour/)”
> * [H] Terrence Tao “[Embracing change and resetting expectations](https://unlocked.microsoft.com/ai-anthology/terence-tao/)”
> * [I] Kevin Buzzard “[Can AI Do Mathematics?](https://www.youtube.com/watch?v=O0F6EFyDA58)”
> * [J] Kevin Buzzard “[Think of a number](https://xenaproject.wordpress.com/2025/01/20/think-of-a-number/)”
> * [K] Kevin Buzzard “[Think of a number: an update](https://xenaproject.wordpress.com/2025/03/16/think-of-a-number-an-update/)”
> * [L] Kevin Buzzard “[Can AI do maths yet? Thoughts from a mathematician](https://xenaproject.wordpress.com/2024/12/22/can-ai-do-maths-yet-thoughts-from-a-mathematician/)”
> * [M] Kevin Buzzard “[Lean in 2024](https://xenaproject.wordpress.com/2024/01/20/lean-in-2024/)”
>
> ## Alternative View
>
> Many machine learning researchers hold an alternative view: neural networks alone can address the challenges of data scarcity and lack of verifiability in informal reasoning, without formal systems. Therefore, there is no consensus on the use of formal systems in mathematical reasoning. Our position paper advocates the use of formal systems and explicitly outlines future milestones, which we believe will significantly benefit the community. In Section 6, we have already provided a discussion of the alternative view. We will revise the paper to include the alternative perspective already in Section 1, to clearly state our position by contrasting it with the alternative view.
>
>
> ##  Clarification on data scarcity
>
> We agree that formal math has less **human-created training data** than informal math (Line 254–261). However, as discussed in Sec. 4.1 and Sec. 2.2 (Line 134–141), formal math enables automatic feedback from proof assistants, which can be used to train models and alleviate the need for human-created data. This is what we meant by “formal math mitigates data scarcity”. Thank you for pointing out the confusion, and we will revise the paper to clarify.
>
>
> ## Missing references
>
> Thank you for the suggestions. We’ll incorporate them in the next revision.

---

> > ### Comment · Reviewer_9EnC · 2025-04-03
> >
> > I thank the authors for the detailed rebuttal. I will have raised my score to a 2: Formal reasoning is well explained to date, and I'm not sure this position paper, while well written and offering good support, will elicit further discussion.

---

> > > ### Author Response · Authors · 2025-04-04
> > >
> > > Thank you for raising the score and for providing valuable feedback for improving the paper!

---

### Decision · Program_Chairs · 2025-04-30

**Decision:**

Accept (spotlight poster)

**Comment:**

The paper argues that formal reasoning methods will be necessary for further development of AI for mathematical reasoning. Three reviewers support the paper (one of them after the rebuttal). The fourth reviewer thinks a paper is a good survey paper but not a good position paper, because the presented position is not new and has already been well discussed in the literature. However, novelty of the position is not a requirement for the Position Paper Track, and a lot of the prior discussion on formal methods is happening outside of the ML community, so this paper would make a meaningful contribution to ICML.